# Different Risk Factors for Erosive Tooth Wear in Rural and Urban Nepal: A National Study

**DOI:** 10.3390/ijerph18157766

**Published:** 2021-07-22

**Authors:** Saujanya Karki, Viivi Alaraudanjoki, Jari Päkkilä, Marja-Liisa Laitala, Vuokko Anttonen

**Affiliations:** 1Research Unit of Oral Health Sciences, University of Oulu, 90014 Oulu, Finland; saujanya.karki@oulu.fi (S.K.); marja-liisa.laitala@oulu.fi (M.-L.L.); vuokko.anttonen@oulu.fi (V.A.); 2Finnish Student Health Services Oulu, 90570 Oulu, Finland; 3Department of Mathematical Sciences, University of Oulu, 90014 Oulu, Finland; jari.pakkila@oulu.fi; 4Medical Research Centre, Oulu University Hospital and University of Oulu, POB 5281, 90014 Oulu, Finland

**Keywords:** erosive tooth wear, obesity, prevalence, socio-demographic factors

## Abstract

Background: Erosive tooth wear (ETW) is of growing concern, but data on ETW among Nepalese children are scarce. The main aim of the study was to investigate the prevalence and severity of ETW among Nepalese schoolchildren. We also aimed to analyse the risk indicators for ETW according to location (rural/urban) and the role of obesity in the risk for ETW. Methods: This national study was conducted among 5–15-year-old Nepalese schoolchildren from different regions. Altogether, 1137 out of 1151 schoolchildren participated in both a clinical examination and a survey. ETW was recorded using the Basic Erosive Wear Examination. Results: The prevalence of ETW was 65%. One-fifth of the examined subjects were in need of preventive or restorative treatment. Living in an urban area and studying in a private school were protective factors for ETW, whereas consuming fruits frequently and using charcoal for tooth cleaning increased the odds for ETW. Central obesity was the strongest risk indicator for ETW among urban residents. Conclusions: ETW of low severity is common among Nepalese children and adolescents. Socio-demographic factors influence the prevalence of ETW in Nepal and there seems to be different factors that play a role in the ETW process according to location of residence.

## 1. Introduction

Erosive tooth wear (ETW) is a multifactorial condition defined as an irreversible loss of dental tissues due to intrinsic or extrinsic acids, exacerbated by mechanical forces [1]. It is of growing concern and has been of interest especially in European countries [2]. According to a recent meta-analysis, the current world-wide prevalence of ETW among children and adolescents is 30% and it has been suggested to be increasing [3,4]. Research on the topic in Asian countries is inconclusive [2], and the prevalence figures vary between low and moderate [5,6,7]. Even though ETW has also gained interest recently in Asian countries, to our knowledge, no studies concerning ETW among Nepalese children and adolescents has been done. The prevalence of tooth wear has, however, been investigated among Nepalese dental patients and it is suggested to be high (60.1%) [8].

The aetiology of ETW is traditionally categorised into extrinsic and intrinsic factors. According to recent findings, high or constant consumption of acidic drinks, fruits, candy and acidic food items is one of the main causes of ETW [9]. In particular, soft (fizzy) drinks and energy drinks can be considered detrimental to teeth due to their low pH and sipping-like intake. Since one of the reasons for childhood obesity is suggested to be the high consumption of sweetened and acidic beverages [10], obesity has also been recently suggested to be associated with ETW [11,12]. However, this potential association needs further studies. Considering the risk factors for ETW altogether, it can be presumed that country- or region-specific habits of living, eating, drinking and even teeth cleaning may play a role in this multifactorial condition. We further infer that in countries such as Nepal, the risk factors for ETW might differ from European ones and there might be some unidentified risk factors.

The main aim of this cross-sectional study was to investigate the prevalence and severity of ETW among Nepalese children and adolescents. We hypothesised that the prevalence of ETW is quite low and severe ETW is rare. Another aim of the investigation was to analyse the role excess weight and obesity in ETW as well as other sociodemographic factors. The hypothesis was that neither BMI nor sociodemographic factors are associated with ETW in Nepal.

## 2. Materials and Methods

This population-based, national cross-sectional study was conducted among 5–6, 12- and 15-year-old Nepalese schoolchildren in April–July 2016. A total of 15 out of 75 districts of Nepal were selected using stratified random sampling (three districts from each administrative developmental region, which also represented the three ecological regions of Nepal). Additionally, all three districts of the Kathmandu valley were included in the study population. One to two schools from each district were conveniently selected.

After power calculation, 1137/1151 schoolchildren participated in both the survey and clinical examinations. The participation response rate was 99%. The power calculation was done using computer software (GPOWER version 3.1) with 95% power using the Mann–Whitney U test between means (difference in the number of decayed primary and permanent teeth, dt/DT = 0.3) with an alpha-type error at 0.05. All 5–6-year-olds accompanied by their parents were included, while randomisation was done for 12- and 15-year-olds based on the sampling fraction. The protocol of the study has been explained in detail earlier [13].

### 2.1. Questionnaire

Sociodemographic information of participants was collected, including date of birth, age, gender (boy/girl), location (urban/rural), school types (public/private) and ethnicity (Brahman and Chhetri/Tarai Madhesi and Muslim/Dalit/Janajati and Newar). A location was considered as urban with access to a health care clinic (government-operated or private) with a dental office and staff within 10 km from the sample site location, and as rural if this access did not exist. This was confirmed after consulting the primary health centre or district health office.

A structured and validated questionnaire was used to collect the information on dietary habits and oral health-related behaviours. Questions on dietary habits included frequencies of intake of fizzy drinks and fruits with options of never, several times a month, once a month, several times a week, every day or several times a day. Similarly, questions on oral health-related behaviours included toothbrushing habits (never, once a month, 2–3 times a month, once a week, 2–6 times a week, once a day or twice or more a day), use of toothpaste (yes or no), use of charcoal for teeth cleaning (yes or no) and use of miswak or a traditional wooden tooth brush or chew stick (yes or no). Parents of 5–6-year-olds were interviewed for filling the questionnaire, while the 12- and 15-year-olds self-administered the questionnaire.

### 2.2. Clinical Oral Examination and Anthropometric Measurements

The clinical oral examination was conducted according to the World Health Organisation criteria and guidelines [14]. ETW was recorded using the BEWE index [15]. The buccal, occlusal and lingual surface were examined in all teeth and the highest score from each sextant was recorded. Finally, a sum score for each individual was created during the analysis.

The details of the quality control and quality assurance were explained in the previous paper [13]. The oral examination was done by three trained and calibrated Nepali dentists in the school premises using an LED headlight, dental mirror and WHO probe. Clinical findings were recorded manually on data-collection sheets by two dental hygienists who acted as enumerators. Training in both theoretical and practical requirements was provided in April 2016 by two senior researchers (M.-L.L. and V.A.) with experience in conducting similar oral health surveys and validation. The training was done using a PowerPoint presentation as well as piloting the clinical examination with five patients. All the five patients had a BEWE score of 0, so Kappa values could not be computed.

The anthropometric measures included were height, weight, waist circumference and hip circumference. The details on the anthropometric measurements and training have been thoroughly explained previously [16].

### 2.3. Statistical Analysis

The manually recorded data were transferred into an electronic database for analyses using SPSS software (IBM SPSS Statistic for Windows, version 24.0, IBM Corp). For analyses, the BEWE sum score of the sextants was categorised as Score 0, Score 1–2 and Score ≥ 3, according to the guidelines [15]. Likewise, dietary habits were also categorised as follows: never (never), seldom (once a month, 2–3 times a month or once a week) and frequent (2–6 times a week, once a day or twice or more than once a day).

Body mass index (BMI) was calculated using the formula: weight divided by the square of the height (kg/m^2^). The waist-to-hip ratio (WHR) was calculated by dividing the waist circumference by the hip circumference, and the waist-to-height ratio (WHtR) by dividing the waist circumference by the height. Participants were categorised according to their BMI as low (underweight) or normal or high (overweight and obese) based on the recent Nepalese growth reference system [16]. A BMI that is below the 12th percentile is considered as low, whereas a BMI above the 88th percentile is considered as overweight, and a BMI above the 98th percentile is considered as obese. Similarly, participants were categorised for central obesity based on their WHR and WHtR, as normal or obese as suggested by the WHO, and the International Diabetes Federation (IDF), respectively.

Distributions were described as proportions and means (SD) and differences between the groups were evaluated using the Chi-square test. For modelling, the BEWE sum score was dichotomised as follows: sum score 0–2 = 0, and sum score ≥ 3 = 1 for analysing the association between the outcome variable and predictors. For modelling, frequency of fruits and fizzy drinks was re-categorised as never and seldom vs. frequently. A generalised mixed model with binary logistic regression was performed for assessing the association. Odds ratios (OR) and 95% confidence intervals (95% CI) were estimated. Model 1 presented the crude odds ratios, while model 2 was adjusted for predictors that were statistically significant during univariate analysis. Model 3 and model 4 were performed for selective cases for those living in an urban and rural location, respectively. Predictors that were statistically significant during univariate analysis were adjusted in both models. The sampling units (school) were considered as random effects in each model. The level of significance was set at *p* ≤ 0.05 for all the statistical tests.

## 3. Results

Among Nepalese schoolchildren, the mean (SD) BEWE sum score was 1.5 (1.4) and the prevalence of ETW (sextant BEWE score > 0) was 65% (5–6 years old: 64%, 12 years old: 64% and 15 years old: 67%). The proportion of those needing preventive or restorative treatment (sum score ≥ 3) was 19%. The mean BEWE scores per sextant did not differ between age groups. The highest BEWE scores per sextants (mean) were found in sextants 4 and 6 (0.5 and 0.6, respectively) and the lowest in sextants 2 and 5 (both 0.1).

Table 1 shows the distribution of the study participants based on their location of residence being urban or rural. There were differences between the groups especially in the frequency of using fizzy drinks, fruits, miswak and fluoride toothpaste. Central obesity was more common among those living in rural areas.

ETW was more common among those living in rural areas, those studying in public schools and those belonging to the Dalit or Tarai Madeshi and Muslim ethnic groups. Similarly, ETW was more prevalent among centrally obese schoolchildren, those eating fruits frequently and the ones using charcoal or miswak for cleaning tooth (Table 2).

Binary logistic regression models revealed that living in an urban area (OR: 0.6; 95% CI: 0.3, 0.9) and studying in private schools (OR: 0.5; 95% CI: 0.3, 0.8) were protective factors for ETW (Table 3). In contrast, consuming fruits frequently compared to never/seldom was associated with ETW (OR: 1.4; 95% CI: 1.0, 2.03). The same was true for using charcoal for tooth cleaning (OR: 2.2, 95% CI: 1.2, 3.9). Similar results were obtained with respect to sociodemographic factors and oral health-related behaviours, when selective case analyses for each age groups (children (5–6-year-olds) and adolescents (12- and 15-year-olds)) were performed. Surprisingly, toothbrushing once daily or less was a protective factor for 5–6-year-olds (OR: 0.32; 95% CI: 0.11–0.93) (Appendix A.).

When selective case analyses were performed for urban and rural areas separately, different associations were observed. Among urban residents, the odds ratio for ETW was high for centrally obese schoolchildren (OR: 2.6; 95% CI 1.5, 4.5) compared to the non-obese ones, whereas being obese was not associated with ETW among rural residents. A similar difference was observed for the association between consumption of fizzy drinks frequently and a high BEWE score (urban area OR: 2.0; 95% CI: 1.1, 3.5; rural area OR: 0.4; 95% CI: 0.2, 0.9). Furthermore, studying in a private school was associated with a low BEWE score (OR: 0.4; 95% CI: 0.2, 0.8); however, it was not significant among rural residents (OR: 0.4; 95% CI: 0.2, 1.0) (Table 3).

## 4. Discussion

This study revealed that two-thirds of the Nepalese children examined had ETW of some degree. One-fifth of the study population was at the medium risk level 15 and severe ETW was rare. A higher BEWE sum score was observed among 5–6-year-olds compared to 15-year-olds. Socio-demographic factors seem to be associated with the risk of ETW in Nepal, since the strongest protective factors were living in an urban area and studying in a private school. Additionally, a high waist–hip ratio and using charcoal for tooth cleaning increased the odds for ETW. Gender was not associated with ETW. Interestingly, selective case analysis showed a varied pattern of association. Among urban residents, a high waist–hip ratio and consuming fizzy drinks frequently increased the odds for ETW. In contrast and surprisingly, among those living in the rural area, consuming fizzy drinks frequently was a protective factor and none of the tested variables were associated with ETW.

To our knowledge, this is the first study to evaluate ETW among children and adolescents in Nepal. One of the strengths of this study is the fairly large study sample with a very high participation rate representing schoolchildren from randomly selected districts of Nepal from different geographic areas and major ethnic groups. The sample can be considered to represent Nepalese children and adolescents. Another strength was the use of calibrated clinical examiners and a validated, modern index for assessing ETW. However, since questionnaires and interviews were used for assessing oral health-related behaviours, it is always possible that there were problems in the children’s or parents’ ability to recall, for example, the consumed food items and drinks. Another limitation is the cross-sectional nature of this study. Given that ETW usually develops over time, habits and lifestyle factors present earlier may be responsible for the current ETW. Among children, this bias can be considered minor compared with adults. It is also known that diagnosing ETW in its early stages is difficult and may cause bias in the study results. It is notable that the inclusion of the 5–6-year-olds who were accompanied by their parents might have led to slight selection bias. Likewise, previous medical history of the participants was not registered; thus, ETW caused by gastric problems, for example, cannot be taken in into account.

The prevalence of ETW in the present study was moderate and is comparable to previous studies from several Asian countries [6,7,17]. However, severe ETW prevalence in the present study is relatively low compared to studies in high-income countries [3,4]. Despite the low proportion of severe ETW, almost half of the present study participants had a BEWE sum score >0. We assume that the trend in ETW may increase in future; at least a similar trend is observed for dental caries, particularly in low- and middle-income countries (LMCs) [18]. The most probable reason for this is the lifestyle and nutritional transitions in LMCs resulting in a remarkable shift towards the consumption of foods and habits that have a potential risk for diseases [19]. This can already be seen among those urban Nepalese residents who consume fizzy drinks, for example, unlike their rural counterparts.

In the present study, almost 1 in 10 had a habit of using traditional Nepalese tooth cleaning materials such as charcoal and miswak (a tooth cleaning twig). The use of these traditional materials is still common in Nepal, especially in the Tarai ecological region [20], which was also included in the present study. Charcoal is known for its abrasive nature on the enamel surface [21], and it is a potential mechanical factor that exacerbates erosive lesions, especially those found in dentine. This can be a reason for the high BEWE sum score among charcoal users in the present study.

One fourth of the 5–6-year-old children were at least in need of preventative measures against ETW (BEWE score ≥ 3) and ETW was more common among them compared to the 15-year-olds in this study. One possible reason can be higher susceptibility of deciduous teeth to erosive wear because of its anatomical structure (thinner enamel layer) [22]. However, an in vitro study contradicts this finding [23], and there might be other factors contributing to the wear of deciduous teeth. One reason for the difference between younger and older age groups could be different oral health-related behaviours amongst the youngest age group and a shift in health behaviours between age groups. Children of the younger age group reported consuming foods with refined sugar frequently and toothbrushing less frequently than older ones [24]. However, 15-year-olds consumed fruits and fizzy drinks and used charcoal for cleaning more frequently than the younger ones.

A lower BEWE sum score was found among those studying in private schools compared to those studying in public schools in this study. This finding is contrary to a population-based study conducted in South Brazil [25], where there was no difference between school types. The same was true for those living in an urban location compared to those living in a rural location. Children from a high socioeconomic status study in private schools in Nepal [26]; among them, the affordability of fluoride toothpaste and seeking oral healthcare personnel are unquestionable. The same might be true for those living in an urban location. We speculate that the protective role of fluoride in erosion may be important here; however, the concentration and efficacy of fluoride in the toothpastes available in Nepal may be questionable and need further investigation. Another speculation can be that these populations receive different amounts of oral health education. All in all, it seems that there are some differences in the ETW-related behaviour according to socio-economic status and these populations might need different kinds of preventive education.

A high odds ratio (OR = 2.5) for ETW was found among the obese urban residents compared to normal waist–hip ratios among urban residents in the present study. Previous studies from high-income countries have also found an association between obesity and erosive wear [11,12]. Soft drinks in particular might be the reason for this, since they are one of the known risk factors for both ETW [23] and obesity [10]. There is evidence showing that the consumption of soft drinks and sugar-sweetened beverages are common among the young and this trend has been increasing globally over the past decades [27]. It has also been reported that lifestyle and nutritional shifts are responsible for the burden of excess weight and obesity in LMICs [28]. As rapid urbanisation in LMICs is leading to a nutritional transition, is can be speculated that ETW might become more prevalent during the next decades.

Surprisingly, frequent consumption of soft drinks was a protective factor among rural residents in this study and in fact the only statistically significantly associated factor with ETW of the analysed variables. It is difficult to give an explanation for this finding, but it might reflect the better socio-economic status of these rural fizzy drink consumers. Maybe these children have better access to fluoride, or they have otherwise better oral health-related habits compared to other rural residents. It is also notable that the proportion of participants consuming fizzy drinks was relatively low in rural Nepal.

Eating fruits frequently is common among urban residents and it was a risk factor for ETW. This finding is consistent with many other studies [29], and just recently, eating fruits daily was found to be the strongest risk factor for tooth wear in a Swedish study [30]. Even though eating fruits is mainly thought of as a healthy habit, this study supports the findings that especially for patients with ETW, the eating of even healthy products should be taken into consideration and eating at regular intervals instead of snacking is highly recommended. The reason why eating fruits was not associated with ETW in rural areas can only be speculated on.

## 5. Conclusions

Altogether, it seems that ETW of low severity is a frequent finding among Nepalese children and adolescents. Socio-demographic factors influence the prevalence of ETW in Nepal and there seems to be different lifestyle factors that play a role in the ETW process according to location of residence. In urban areas, central obesity and fizzy drink consumption are associated with ETW, reflecting the transition towards high-income countries and eventually bringing new challenges to the dental field in Nepal. In rural areas, there seems to be unidentified factors causing or exacerbating ETW and further research of the aetiology of ETW is needed.

## Figures and Tables

**Table 1 ijerph-18-07766-t001:** Distribution (%) of the participants according to their location of residence stratified by age, gender, school type, ethnic group, fizzy drinks consumption, fruits consumption, toothbrushing frequency, use of toothpaste, use of charcoal, use of miswak, body mass index, waist–hip ratio, waist–height ratio and BEWE score.

Characteristics	Location (%)		*p*-Value
	Urban (n = 606)	Rural (n = 531)	
**Age**			
5–6 years old	28.5	31.5	0.170
12 years old	35.3	37.7	
15 years old	36.1	30.9	
Gender			
Boys	50.8	54.4	0.225
Girls	49.2	45.6	
School types			
Public	50.7	72.3	<0.001
Private	49.3	27.7	
Ethnic group			
Brahman and Chhetri	47.7	49.9	<0.001
Tarai Madeshi and Muslim	12.4	0.9	
Dalit	8.7	9.8	
Janajati and Newar	31.2	39.4	
Fizzy drinks consumption			
Never	8.6	14.4	<0.001
Seldom	61.6	70.5	
Frequent	29.8	15.1	
Fruits consumption			
Never	1.2	2.6	<0.001
Seldom	41.9	63.3	
Frequent	56.9	34.1	
Toothbrushing frequency			
Once daily or less	79.1	80.6	0.569
Twice daily or more	20.9	19.4	
Use of toothpaste			
Yes	99.3	97.4	0.012
No	0.7	2.6	
Use of charcoal			
Yes	6.0	8.3	0.159
No	94.0	91.7	
Use of miswak			
Yes	10.2	6.3	0.027
No	89.8	93.7	
Body mass index			
Low	11.6	13.8	0.355
Normal	79.2	78.7	
High	9.3	7.5	
Waist–Hip ratio			
Normal	71.0	62.0	0.001
Obese	29.0	38.0	
Waist–Height ratio			
Normal	93.9	87.6	<0.001
Obese	6.1	12.4	
BEWE score			
Score 0	36.6	33.5	0.273
Score ≥1	63.4	66.5	

**Table 2 ijerph-18-07766-t002:** Distribution (%) of the participants according to the Basic Erosive Wear Examination (BEWE) sum scores stratified by age, gender, location, school types, ethnic group, fizzy drinks consumption, fruits consumption, toothbrushing frequency, use of toothpaste, use of charcoal, use of miswak, body mass index, waist–hip ratio and waist–height ratio.

Characteristics	BEWE Sum Scores (%)			*p*-Value
	Score 0 (n = 400)	Score 1–2 (n = 517)	Score ≥ 3 (n = 220)	
Age				
5–6 years old	36.2	38.2	25.6	<0.001
12 years old	36.5	44.4	19.1	
15 years old	32.9	53.0	14.1	
Gender				
Boys	36.2	44.4	19.4	0.710
Girls	34.1	46.7	19.3	
Location				
Urban	36.6	49.3	14.0	<0.001
Rural	33.5	41.1	25.4	
School type				
Public	31.5	45.9	22.6	<0.001
Private	40.8	44.8	14.3	
Ethnic group				
Brahman and Chhetri	33.9	44.8	21.3	0.035
Tarai Madeshi and Muslim	26.3	53.8	20.0	
Dalit	28.6	47.6	23.8	
Janajati and Newar	40.5	44.2	15.3	
Fizzy drinks consumption				
Never	30.2	49.1	20.7	0.494
Seldom	35.6	46.6	17.8	
Frequent	39.3	42.1	18.6	
Fruits consumption				
Never	21.1	63.3	15.8	0.006
Seldom	31.9	50.5	17.6	
Frequent	40.7	40.1	19.1	
Toothbrushing frequency				
Once daily or less	35.7	45.4	18.8	0.660
Twice daily or more	36.2	47.6	16.2	
Use of toothpaste				
Yes	35.7	46.1	18.2	0.487
No	43.8	31.3	25.0	
Use of charcoal				
Yes	21.9	45.2	32.9	0.001
No	36.8	46.0	17.2	
Use of miswak				
Yes	21.6	58.0	20.5	0.013
No	37.2	44.7	18.1	
Body mass index				
Low	33.6	46.2	20.3	0.830
Normal	34.8	45.8	19.4	
High	40.6	41.7	17.7	
Waist–Hip ratio				
Normal	35.4	48.4	16.2	<0.001
Obese	34.7	39.7	25.7	
Waist–Height ratio				
Normal	35.8	45.4	18.9	0.268
Obese	29.1	46.6	24.3	
Total	35.2	45.5	19.3	

**Table 3 ijerph-18-07766-t003:** Association between erosive tooth wear (sum score 0–2/≥3) and explanatory variables (age, gender, location, school types, ethnic group, fizzy drinks consumption, fruits consumption, toothbrushing frequency, use of toothpaste, use of charcoal, use of miswak, body mass index, waist–hip ratio and waist–height ratio) using logistic regression models.

Explanatory Variables	Model 1 UOR (95% CI)	Model 2 OR (95% CI)	Model 3 OR (95% CI)	Model 4 OR (95% CI)
Age				
5–6 years old	2.30 (1.54–3.45) **	2.01 (1.18–3.45) *	-	-
12 years old	1.36 (0.92–2.02)	1.33 (0.89–1.99)	-	-
15 years old	1	1	-	-
Gender				
Boys	0.96 (0.70–1.30)	-	-	-
Girls	1	-	-	-
Location				
Urban	0.53 (0.32–0.87) *	0.57 (0.34–0.93) *	-	-
Rural	1	1	-	-
School types				
Private	0.53 (0.30–0.93) *	0.49 (0.29–0.83) *	0.38 (0.18–0.80) *	0.43 (0.18–1.04)
Public	1	1	1	1
Ethnic group				
Brahman and Chhetri	1.10 (0.72–1.68)	1.13 (0.74–1.74)	0.77 (0.40–1.43)	2.04 (1.10–3.78) *
Tarai Madeshi and Muslim	0.93 (0.44–1.97)	1.07 (0.50–2.31)	0.95 (0.39–2.31)	1.34 (0.13–13.96)
Dalit	1.34 (0.77–2.34)	1.20 (0.66–2.18)	1.55 (0.65–3.69)	1.05 (0.44–2.52)
Janajati and Newar	1	1	1	1
Fizzy drinks consumption				
Frequent	1.17 (0.80–1.73)	-	1.98 (1.15–3.49) *	0.41 (0.19–0.89) *
Never and seldom	1	-	1	1
Fruits consumption				
Frequent	1.34 (0.96–1.88)	1.43 (1.01–2.03) *	2.01 (1.11–3.63) *	0.93 (0.57–1.53)
Never and seldom	1	1	1	1
Toothbrushing frequency				
Once daily or less	1.18 (0.77–1.80)	-	-	-
Twice daily or more	1	-	-	-
Use of toothpaste				
Yes	0.95 (0.29–3.12)	-	0.65 (0.06–7.55)	1.25 (0.30–5.15)
No	1	-	1	1
Use of charcoal				
Yes	1.88 (1.08–3.27) *	2.16 (1.20–3.85) *	-	-
No	1	1	-	-
Use of miswak				
Yes	1.04 (0.59–1.84)	1.07 (0.58–1.96)	1.12 (0.50–2.49)	1.20 (0.49–2.97)
No	1	1	1	1
Body mass index				
Low	0.78 (0.49–1.24)	-	-	-
High	1.09 (0.62–1.93)	-	-	-
Normal	1	-	-	-
Waist–Hip ratio				
Obese	1.76 (1.28–2.42) *	1.35 (0.89–2.06)	2.56 (1.47–4.47) *	1.02 (0.60–1.73)
Normal	1	1	1	1
Waist–Height ratio				
Obese	1.37 (0.82–2.29)	-	1.96 (0.77–4.96)	1.38 (0.58–3.30)
Normal	1	-	1	1

* *p* < 0.05; ** *p* < 0.001; UOR: unadjusted odds ratio; 95% CI: 95% confidence interval; OR: odds ratio; Model 2 was adjusted for age, location, school types, ethnic groups, fruits consumption, use of charcoal, use of miswak and waist–hip ratio; Model 3 was performed for participants living in an urban area, and was adjusted for school types, ethnic groups, fizzy drinks consumption, fruits consumption, use of toothpaste, use of miswak, waist–hip ratio and waist–height ratio; Model 4 was performed for participants living in a rural area, and was adjusted for school types, ethnic groups, fizzy drinks consumption, fruits consumption, use of toothpaste, use of miswak, waist–hip ratio and waist–height ratio.

## Data Availability

Data are available upon request.

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
