# Peer review of "Different Risk Factors for Erosive Tooth Wear in Rural and Urban Nepal: A National Study"

_ijerph, 2021, doi:10.3390/ijerph18157766_

Round 1

Reviewer 1 Report

This manuscript reports on the situation of erosive tooth wear among the children in Nepal. The study seems to have been properly conducted but the reporting should be improved.  The major problem lies in pooling the situation in the primary dentition of the 5-6 year-old children with that of the permanent dentition of the 12- and 15-year-old children. Since the primary and permanent dentition would have different resistance to tooth wear and the dietary practice of preschool children and that of the older children would be quite different, the risk factors for tooth wear would likely to be different. I would suggest the authors to analyze the data and report the results separately for the 5-6-year-old children and for the adolescents. 

Author Response

This manuscript reports on the situation of erosive tooth wear among the children in Nepal. The study seems to have been properly conducted but the reporting should be improved.  The major problem lies in pooling the situation in the primary dentition of the 5-6 year-old children with that of the permanent dentition of the 12- and 15-year-old children. Since the primary and permanent dentition would have different resistance to tooth wear and the dietary practice of preschool children and that of the older children would be quite different, the risk factors for tooth wear would likely to be different. I would suggest the authors to analyze the data and report the results separately for the 5-6-year-old children and for the adolescents. 

Thank you for your comments.

We have discussed this in our previous version (Discussion section, 5th paragraph). In addition, we have now provided new analyses for 5-6-year-olds and 12- and 15-year-olds as a supplement file. Results are interpreted in page 6 line 163-167.

Reviewer 2 Report

The study is interesting and it is well-written and performed. In my opinion the study just needs careful reading for English language corrections.

For example, the first sentence of the Introduction is: “Erosive tooth wear (ETW) is a multifactorial condition and defined as an irreversible...”. This should be: “Erosive tooth wear (ETW) is a multifactorial condition and is defined as an irreversible...”.

Later on, at line 32: “According to the recent meta-analysis…”. This should be changed to “According to a recent meta-analysis…”.

Please read the entire manuscript carefully and address such issues.

Author Response

The study is interesting, and it is well-written and performed. In my opinion the study just needs careful reading for English language corrections.

For example, the first sentence of the Introduction is: “Erosive tooth wear (ETW) is a multifactorial condition and defined as an irreversible...”. This should be: “Erosive tooth wear (ETW) is a multifactorial condition and is defined as an irreversible...”.

Later on, at line 32: “According to the recent meta-analysis…”. This should be changed to “According to a recent meta-analysis…”.

Please read the entire manuscript carefully and address such issues.

Thank you for your comments, manuscript has now undergone English language proof reading.

Reviewer 3 Report

The authors present a properly structured well written manuscript on the prevalence of erosive tooth wear in Nepalese children. The study is population based, cross-sectional on a large cohort of two age groups amongst children. The study provides no new insights into ETW aspects, but is a unique study on the population of Nepal, which makes it valuable and interesting. 

I do have several comments on the manuscripts:

Line 41: "food items" is to vague, remove or replace.

Lines 46-48: try replacing one of the "speculates" with a synonym.

Materials and methods: several times you quote previously published data regarding this publication (protocol, etc.). Though this is acceptable, I would suggest an overview of these areas, so the reader of the current paper will have an acceptable grasp on the methodology. A figure of the algorithm could be helpful. 

Discussion: very well written. Just try to avoid using the term "fizzy". It is proper to say "sparkling" or "soft".

Conclusion: well formulated and precise. 

In conclusion, the authors present a well drafted manuscript with abundant relevant literature. Limitations are properly addressed. 

Author Response

The authors present a properly structured well written manuscript on the prevalence of erosive tooth wear in Nepalese children. The study is population based, cross-sectional on a large cohort of two age groups amongst children. The study provides no new insights into ETW aspects, but is a unique study on the population of Nepal, which makes it valuable and interesting. 

 Thank you for your comments.

I do have several comments on the manuscripts:

Line 41: "food items" is to vague, remove or replace.

We have now replaced “food items” with the “acidic food items”.

Lines 46-48: try replacing one of the "speculates" with a synonym.

It is now replaced with “presumed” and “inference”.

Materials and methods: several times you quote previously published data regarding this publication (protocol, etc.). Though this is acceptable, I would suggest an overview of these areas, so the reader of the current paper will have an acceptable grasp on the methodology. A figure of the algorithm could be helpful. 

Thank you for your comment. To avoid the repetition to have referred to our previous papers that clearly describes the training and calibration as well as method of recordings. 

We would like to stick to the current form.

Discussion: very well written. Just try to avoid using the term "fizzy". It is proper to say "sparkling" or "soft".

We have now replaced the term “fizzy” with “soft” in the discussion section (wherever possible). However, we will be consistent with the term “fizzy drinks” in our methods and results section as the original questionnaire required to do so.

Conclusion: well formulated and precise. 

In conclusion, the authors present a well drafted manuscript with abundant relevant literature. Limitations are properly addressed. 

Thank you for your comment. We appreciate them.